# Biofilms: Novel Strategies Based on Antimicrobial Peptides

**DOI:** 10.3390/pharmaceutics11070322

**Published:** 2019-07-10

**Authors:** Emilia Galdiero, Lucia Lombardi, Annarita Falanga, Giovanni Libralato, Marco Guida, Rosa Carotenuto

**Affiliations:** 1Department of Biology, University of Naples Federico II, Via Cinthia, 80126 Naples, Italy; 2Department of Pharmacy, University of Naples Federico II, Via Mezzocannone 16, 80134 Naples, Italy; 3Department of Agricultural Sciences, University of Naples Federico II, Via Università 100, 80055 Portici, Italy

**Keywords:** anti-biofilms peptides, biofilm in vitro, biofilm in vivo, nanotechnology

## Abstract

The problem of drug resistance is very worrying and ever increasing. Resistance is due not only to the reckless use of antibiotics but also to the fact that pathogens are able to adapt to different conditions and develop self-defense mechanisms such as living in biofilms; altogether these issues make the search for alternative drugs a real challenge. Antimicrobial peptides appear as promising alternatives but they have disadvantages that do not make them easily applicable in the medical field; thus many researches look for solutions to overcome the disadvantages and ensure that the advantages can be exploited. This review describes the biofilm characteristics and identifies the key features that antimicrobial peptides should have. Recalcitrant bacterial infections caused by the most obstinate bacterial species should be treated with a strategy to combine conventional peptides functionalized with nano-tools. This approach could effectively disrupt high density infections caused by biofilms. Moreover, the importance of using in vivo non mammalian models for biofilm studies is described. In particular, here we analyze the use of amphibians as a model to substitute the rodent model.

## 1. Introduction

The growing number of nosocomial infections associated to the emergence of microorganisms resistant to conventional antibiotics has been recognized as one of the major concerns affecting modern healthcare.

It has been estimated that less than 0.1% of the total microbial population is in the planktonic mode of growth (isolated bacteria cells). Instead, the majority of the bacteria are organized in more complex structures known as biofilms. Accordingly, more than 80% of all bacterial infections are due to microorganisms in their biofilm mode of growth [1].

Biofilms can form in any host tissue causing pulmonary infections in patients with cystic fibrosis, infections in diabetic patients, recurrent urinary tract infections, or colonization of abiotic surfaces such as central venous or urinary catheters, heart valves, intrauterine devices, and dental implants. Formation of biofilms is not the only aspect responsible of chronic infections, which are difficult to diagnose and treat with conventional antibiotic treatments, but also current antimicrobial therapy practices have produced a rapid growth of drug-resistant infections [2,3]. Moreover, mass use (both for humans and animals) and easy access of the antibiotics has led to further development of resistance, making the problem even more serious for our society from both an economical and health perspective.

In their biofilm form, bacteria adhere on a surface and grow while being surrounded by a self-produced matrix of extracellular polymeric substances (EPS) such as polysaccharides, proteins, and extracellular DNA (eDNA) that protect biofilm inhabitants against the host immune system and antimicrobial treatments [2,4]. 

Defeating bacteria within the biofilm matrix is challenging not only because the matrix functions as a protecting screen but also because bacteria assume a dormant state (persisters) which can significantly compromise the efficacy of currently available antibiotics, since their mechanisms of action interfere with bacterial metabolism or proliferation. In their biofilm state, bacteria can also be resistant towards the host’s innate and adaptive immune and inflammatory defense systems. Hence, a global interest is directed towards the need to solve the problem of bacterial resistance.

Biofilms are less susceptible to antimicrobial treatment and to the host immune system than planktonic cells, thus it could be important to develop an ideal strategy to combat chronic biofilm related infections strengthening the search for new antimicrobials.

## 2. Understanding Biofilm Properties

Properties of biofilm lifestyle have to be considered when peptide strategies are proposed for the prevention of its formation or in general to design promising anti-biofilm agents. Biofilms develop in three phases: initial adhesion, maturation, and dispersion. The first phase is mediated by some host molecules such as albumin, lipids, complements, fibronectins, and by the interactions of some bacterial surface structures such as pili, fimbriae, but is also influenced by the environment, the micro-organisms properties, the material surfaces, and the flow conditions [5,6].

The maturation is especially associated with the EPS production, which allows the stabilization of the biofilm. The last phase, the dispersion, corresponds to the release of cells that colonize new sites. Mature biofilms change microenvironment conditions such as pH, oxygen level, and weaken the activity of antibiotics as well as create particular conditions to induce the state of dormant and non-dividing cells called “persisters” that are believed to play an important role in resistance.

The formation of *P. aeruginosa* biofilms is very frequent in patients with cystic fibrosis, while *S. aureus* is frequently correlated with wounds and the multi-species biofilms with urinary catheter or cardiovascular devices.

Recently, it has been reported that the diseases caused by *Candida albicans* and non-albicans *Candida* (NAC) spp., such as *C. glabrata, C. tropicalis, C. krusei, C. dubliensis*, or *C. parapsilosis* are associated with biofilm growth and, in late stages, their biofilms are more resistant to antifungal drugs. *Candida* biofilm formation is characterized by three development phases which are the early adhesion phase (0–11 h), intermediate phase (12–30 h), and maturation phase (31–72 h). In order to colonize and form biofilms, cells must attach to the surface or host cells becoming more resistant; new formulations, new agents, and combination therapies are urgently needed to prevent adherence and biofilm development [7].

An effective anti-biofilm agent should act in different environmental niches in order to reach cells in different growth rate, and also persisters. For very resistant biofilms, a combination of an agent acting on metabolic active cells and another one able to act on persisters could represent a valid strategy.

EPS is the major component of biofilms; it produces an appropriate environment for the bacteria, favors the adhesion cell-to-cell and cell-to-surface but also constitutes a barrier that interferes with the action of antimicrobial agents decreasing their bioactive concentrations. Therefore, good anti-biofilm agents should be able to penetrate the matrix or inhibit the accumulation on the surface. As cells in biofilms can communicate through secretion of molecules known as auto-inducers that control the quorum sensing (QS), a promising anti-biofilm agent could interfere with QS signals favoring biofilm inhibition or dispersion.

A drug that exhibits all these anti-biofilm properties together is still not available, even if many studies are devoted to design antimicrobial peptides (AMPs) which can display a combination of these features [8,9].

## 3. Antimicrobial Peptides as Anti-Biofilm Agents

Antimicrobial peptides [10,11,12,13] are a promising class of compounds, widely distributed in nature, mainly as a part of innate immunity of organisms and are currently receiving considerable attention as a potential alternative to conventional antibiotics to prevent and treat bacterial infections [3].

They have a broad spectrum activity against many microbes ranging from viruses to parasites, and are also less susceptible to evolved antibacterial resistance because of their fast mechanism of action, rapid protease digestion with low bioavailability, and peptide aggregation [14]. Moreover, they exhibit a low probability to develop antimicrobial resistance likely thanks to their membrane-associated activity. Many AMPs are positively charged, amphiphilic molecules that kill bacteria through membrane disruption and/or pore formation, but also other modes of action are envisioned, such as penetration into the bacterial cytoplasm and inhibition of proteins, cell wall, or enzymes synthesis [15]. Many studies also showed that AMPs are effective against multi-drug resistant bacteria and fungi not only because they act disrupting membranes due to the electrostatic negatively charged bacterial surfaces, but also interfering with metabolic processes such as inhibition of cell wall, nucleic acid or protein biosynthesis rendering more difficult for microorganism to develop resistance.

However, positive charges although being essential for antimicrobial action also represent an impediment to AMPs penetration into biofilms [16]. The main reason lies in the fact that polysaccharide (EPS) of the biofilm matrix is negatively charged and can trap positive AMPs preventing them from displaying their action against bacteria. Moreover, many AMPs are subject to hydrolytic and proteolytic breakdown.

Several studies have described AMPs and how to modify their sequence in order to obtain novel antibiofilm agents with increased activity and proteolytic stability. Among them, LL-37 has shown an activity against *S. aureus* bacteria as well as in preventing action against biofilm formation. However, despite the high potency of this peptide, the use of LL-37 is limited because of its poor stability; indeed, it is easily cleaved by endogenous enzymes highly present in the gut (trypsin and pepsin), pancreas (elastace) and serum (plasmin). Further, also enzymes secreted by microorganisms including *S. aureus* (aureolysin, V8 protease) have shown the ability to cleave LL-37. Another disadvantage is associated with non-specific cytotoxicity (e.g., leukocyte, T-cell, and red blood cell viability). Shorter derivatives of LL-37 can overcome some of these limitations by improving stability, reducing toxicities, and lowering production costs [17,18,19,20,21,22,23]. de la Fuente-Nunez et al. [24] identified the short cationic peptide 1037 with a weak antimicrobial activity on Gram-positive and Gram-negative bacteria but with a good anti- biofilm activity with death of cells. Moreover, they showed a decreased expression of a variety of genes involved in biofilm formation for *P. aeruginosa*.

In a recent study, de Alteriis at al. demonstrated that a membrane-penetrating peptide gH625 and its analogue gH625-GCGKKKK were effective both on planktonic cells and *C. tropicalis/Serratia marcenscens* and *C. tropicalis/S. aureus* polymicrobial biofilms [25]. They showed an eradication activity stronger for the modified analogue, also at concentration lower than their minimal inhibitory concentration (MIC) on planktonic cells.

Many AMPs and peptido-mimetics present an action against microbial biofilms. They act in the first phase of biofilm formation and display a preventive action, or disperse existing ones, reduce biomass or kill microbial cells within biofilms as schematized in Figure 1. Some peptides perform different antimicrobial actions according to the concentration; for instance, at certain concentrations they are able to limit biofilm formation without affecting planktonic growth of the bacteria [26].

In vivo studies demonstrated that biofilms are also capable of escaping host immune responses leading to an intense host pro-inflammatory response, which contributes to tissue damage and a lower action of macrophages [27]. Another required property for anti-biofilm agents is the capability of balancing the pro-inflammatory and anti-inflammatory immune cell responses.

AMPs can be divided mainly into two groups: natural and synthetic. Natural AMPs are part of the natural anti-inflammatory and antimicrobial system and display a synergistic effect with antibiotics; they represent a source of inspiration for the design of synthetic AMPs and their main disadvantage is the high instability due to the rapid degradation and easy cleavage by proteases. To increase the potential of therapeutic applications of AMPs, novel chemically modified analogues have been synthetized to enhance their pharmacological properties [28,29,30,31].

Most AMPs are toxic to bacteria but not to eukaryotes owing to the different membrane organization of bacteria and multicellular organisms [23,32]. Moreover, they present different and faster killing mechanisms compared to conventional antibiotics because exploiting the disruption and permeabilization of microbial cell membranes.

Among natural AMPs, those derived from amphibians certainly hold great promise; in fact, the amphibians present a surprisingly diverse population of antimicrobial peptides [33]. New AMPs are isolated from amphibian secretions, which are mainly produced in their skin granular glands and are released following stress or injury [34,35].They have received great attention in recent years and as a matter of fact, more than a thousand of amphibian peptides are deposited in Antimicrobial Peptide Database [36,37,38]. Because their innate resistance to bacterial infections the amphibians have been studied since the 1930s [39] and in 1987, for the first time, Zasloff extracted two potent nonhemolytic antimicrobial peptides from *X. laevis* skin [40]. AMPs are frequently isolated from amphibians as a mold to produce synthetic analogues to fight biofilms.

Here we list some amphibian AMPs isolated in recent years.

Li et al. [41] isolated a small anionic antibacterial peptide from *X. laevis* secrections (XLAsp-P1). XLAsp-P1 showed high activity in vitro against a great number of Gram-positive and Gram-negative bacteria with MIC starting at 10 μg/mL. Moreover, it shows only 6.2% of hemolytic activity when used at 64 μg/mL. This peptide acts on the membrane.

Wu et al. [42] used the skin secretion of *Litoria infrafrenata* to discover peptides with therapeutic potential; they cloned a frenatin gene and identified the peptide frenatin 4.1 and a post-translation modified peptide, frenatin 4.2. Moreover, they designed analogues that exhibited enhanced antimicrobial activities as frenatin 4.2a.

Huang et al. [43] extracted from the mucus of *Phyllomedusa hypochondrialis*, Dermaseptin-PH, and investigated its antimicrobial activity. Dermaseptin-PH inhibited the growth of both Gram-negative and Gram-positive bacteria and of *C. albicans*. Liu et al. [44] showed that phylloseptin-PTa and phylloseptin-PHa isolated from *Phyllomedusa tarsius* and *Phyllomedusa hypochondrialis* have antimicrobial activities against Gram-positive bacteria (such as *Staphylococcus aureus* and *Enterococcus faecalis*) and methicillin-resistant *Staphylococcus aureus*.

Sang et al. [45] have extracted from the secretion of *Pelophylax kl. esculentus* skin the temporin-PE, a potent AMP (Table 1). Modifications of this peptide introducing short sequences, as TAT, at C-terminus or modifications of FLP motif produced variations of its activity.

Pei et al. [46] isolated andricin B from the blood of *A. davidianus* by a new method which implied the magnetic liposome adsorption combined with reversed-phase high-performance liquid chromatography. Andricin B is active against all bacteria tested and some fungi. The MICs were in the range 8–64 μg/mL. Moreover, the hemolytic tests indicated that andricin B is safe at the MICs at which it inhibits the growth of *S. aureus.*

Yuan et al. [47] isolated Japonicin-2LF from the skin mucus of *Limnonectes fujianensis* through the combination of cDNA cloning and MS/MS sequencing. The activity of this peptide was evaluated with different methods. They used the *Galleria mellonella* infection model to assess the efficacy of Japonicin-2LF. This peptide showed great antimicrobial activity against *S. aureus* and MRSA, killing the bacteria via membrane permeabilization. Japonicin-2LF acts through inhibition and removal of biofilms, especially against MRSA biofilm. In vivo studies indicated that Japonicin-2LF is a potential candidate to control the MRSA infection in cystic fibrosis patients.

Most anti-biofilm peptides work at concentrations equal or higher than their MIC but there are some peptides that showed their action also at concentration much lower than MIC interfering with the biofilm life-style and displaying that the optimal antimicrobial activity against planktonic cells does not necessarily coincide with an optimal anti-biofilm effect. In other words, anti-biofilm activity is not necessarily correlated to antimicrobial activity.

Generally, the minimal biofilm inhibiting concentration (MBIC) of AMPs is much lower than minimal inhibitory concentration of bacterial growth (MIC). Sometimes they prevent the intracellular accumulation of secondary messenger nucleotide molecules (ppGpp, c-d-GMP) which are involved in many metabolic steps such as colonization, adhesion, and communication in Gram-positive and Gram-negative bacteria [14]. Extracellular DNA (e-DNA) is a major component of bacteria and fungi biofilm. It is known that in biofilm e-DNA acts as intracellular connector to maintain the biofilm architecture, but it is also known that it may favor the exchange of a pool of genes interested in acquisition of virulence or resistance in bacteria forming biofilm, its release is under control of QS signals, and, finally, it could be a nutrient source or a cation chelator. Consequently, a possible anti-biofilm strategy could be directed on e-DNA composition.

Moreover, EPS also plays an essential role in biofilm life-style allowing the initial colonization and the accumulation of nutrients and stabilization of enzymes. It has been demonstrated that EPS promotes the self-association of cationic peptides allowing a better interaction with membrane bilayers but, on the other side, EPS entraps AMPs before they can reach bacterial targets so that they constitute a protective environment against the innate host defense during an infection [26,48]. For a better strategy, a good anti-biofilm peptide should carry the right balance between antimicrobial activity and EPS permeability.

Despite their promising properties, AMPs are not used in clinical settings because of their potential toxicity at efficacious doses, instability in human fluids, as well as ecological toxicity, which is under analysis.

## 4. Nanoparticles Coated with AMPs

Nanotechnology through the combination of AMPs and nanoparticles (NPs) could improve the antimicrobial action of the peptides but also overcome their disadvantages [49,50,51]. Nanotechnology offers a number of promising opportunities to develop antimicrobial nanosystems that break the biofilm barrier and penetrate over significantly larger distances into biofilm, killing its microorganisms more than conventional antibiotics or AMPs.

Both the multidrug resistance and protective character of biofilms could make common antibiotics thousands-fold less potent. Despite the initial enthusiasm for AMPs as alternative candidates to antibiotics, disadvantages have limited their development and clinical use. Thanks to nanotechnology, many nanomaterials (1–100 nm) have been developed with powerful capability of delivering drugs. Among them, NPs are widely exploited in different fields. According to the application, the NP composition can vary and for biomedical applications, various biocompatible materials have been exploited. Many NPs have been demonstrated to have anti-biofilm activities by themselves, but nonetheless, they can carry antimicrobial agents such as AMPs and show enhanced properties.

Silver NPs, gold NPs, quantum dots (QD) NPs have anti-biofilm activities even though their toxicity remain a serious issue which prevents their full application.

Functionalization of NPs surface with AMPs is attracting growing interest for three main reasons: (i) NPs can facilitate delivery; (ii) NPs make peptides more resistant towards degradation; (iii) peptides form a protecting/coating shell on the NPs, which can reduce their toxic effect. Thus, when conjugated, peptides and NPs combine the antimicrobial activity of both components and are also able to overcome their drawbacks. Thus, functionalization of NPs with AMPs might represent a novel approach to overcome some problems related to the limited clinical use of AMPs, as reported in Figure 2.

Covalent immobilization of AMPs and their analogues is widely exploited. Although being a promising strategy, covalent immobilization appears to produce new properties and functions that are not owned by their building blocks [52], which may influence their interaction with cell membranes and their activity. As a consequence, the antimicrobial activity may be affected because the charge distribution will be different from that of single peptide molecules.

A novel strategy to develop nanosystems coated with AMPs is represented by the use of self-assembled peptides for the obtainment of on demand supramolecular nanostructures (nanotubes, nanobelts, fibrils, nanovesicles, gels, and nanocages), which can disassemble upon contact with the bacteria and release the AMP. Recently, a novel versatile platform was developed in our laboratory to immobilize an analogue of the marine antimicrobial peptide myxinidin (WMR) on a peptide based biomaterial [53]; the AMP sequence was located on the periphery of the nanofiber, and was able to increase their effective local concentration compared to soluble peptides and was the driving force for improved antibacterial activity [33,54,55,56]. The presence of WMR on self-assembled nanostructures improved anti-biofilm activity against the Gram-negative bacterium *P. aeruginosa* and the fungus *C. albicans*; the fibers were able both to inhibit the biofilm formation and to eradicate pre-formed biofilms [53,57].

Almaaytah et al. showed that RBRBR-CS-NPs, a polymeric NPs functionalized with RBRBR, cationic ultrashort AMPs, were active against wild-type and the multidrug-resistant clinical isolated strains of Gram-positive bacteria compared to the peptide-unloaded nanoparticles [58]. The antimicrobial effect is very powerful also against clinical isolates of the resistant strains of *S. aureus* compared to free RBRBR.

Boden et al. [59] studied a highly cationic peptide PuroA (FPVTWRWWKWWKG-NH_2_), derived from the tryptophan-rich domain of wheat puro indoline proteins, chosen because of its broad spectrum of antimicrobial activity [16]; the peptide was conjugated with a colloid BCC layer as a suitable platform with inherent nano-/microstructures and topographies for covalent immobilization of AMPs. The results showed a substantial increase of the anti-biofilm properties of PuroA-modified BCCs compared to the activity of unbound PuroA demonstrating also that changes in topography affect bacterial attachment profiles and growth.

To date, comprehensive studies of AuNPs on bacteria have rarely been carried out and very little is known about the biological effects of AuNPs conjugated to AMPs [60,61]. Casciaro et al. identified a derivative of the frog skin AMP esculentin-1a, esculentin-1a(1-21)NH_2_[Esc(1-21)], GIFSKLAGKKIKNLLISGLKG-NH_2_, corresponding to the first 20 residues of esculentin-1a, with a strong activity against both free-living and sessile forms of either reference or clinical isolates of *P. aeruginosa* [61]. Esc(1-21)-coated AuNPs were tested for their antibiofilm activity; when used against the sessile form of *P. aeruginosa*, 50% of the biofilm cells was killed within 2 h treatment at a peptide concentration of 0.171 M which was only two-fold higher than the MBC50, and about 17-fold lower than the MBEC 50 of the free peptide. Therefore, the frog skin AMP esculentin-1a conjugated to AuNPs via PEG linker, demonstrated a remarkably improved antibacterial activity compared to the free peptide without being toxic to human cells. This is likely due to the significant quantity of Esc(1-21) present on the surface of AuNPs@PEG, regardless of its orientation, and to the high concentration of AuNPs@Esc(1-21) on the bacterial surface as well as to the prolonged peptide bioavailability, being less accessible to bacterial proteases. The functionalized AuNPs@Esc(1-21) also presented a membrane perturbing activity as a plausible mechanism of bacterial killing.

Galdiero et al. [62] reported that the antimicrobial activity of QDs is enhanced by the functionalization with the antimicrobial peptide indolicidin (QDs-Ind) against *S. aureus* (ATCC 6538), *P. aeruginosa* (ATCC 1025), *E. coli* (ATCC 11229), and *Klebsiella pneumoniae* (ATCC 10031), while the ecotoxicity decreases compared to the QDs and indolicidin separately using an ecotoxicological battery of test systems and indicators able to detect different effects using a variety of end points [62].

de Alteriis et al. [63] studied the efficacy in vitro of a nanocomplex of AuNPs coated with indolicidin (AuNPs–indolicidin) in both preventing cell adhesion and eradicating the developed biofilms of some *Candida* clinical isolates and ATCC strains showing an enhanced indolicidin activity likely related to an enhanced stability of the nano-complex towards protease degradation occurring in the matrix and/or inside cells.

Atefyekta et al. [64] showed that RRP9W4N covalently attached on the elastin-like polypeptide (ELP) films had anti-biofilm activity toward *S. epidermidis, S. aureus*, and *P. aeruginosa*, and moreover, the stability of covalently attached AMP in human serum was highly increased showing high activity up to 24 h [18].

Mohid et al. [65] designed two hybrid AMPs, KG18 and VR18, from the parent peptides VG13P and WR17 [19]. These chimeric peptides display increased and broad-spectrum activity compared to their parents; moreover, conjugation of ultra-small tungsten disulfide (WS2) QDs to these peptides was performed to further increase their antimicrobial activity. They proved that these AMP NPs have a potent antimicrobial effect against both planktonic and biofilm-forming pathogens due to nanoparticles that enhance membrane lysis and antimicrobial/antibiofilm activities. The combination of non-toxic ultra-small QDs with potent and selective AMPs represents a promising tool for control of infections caused by bacterial and fungal pathogens.

Barbosa et al. [66] evaluated the antimicrobial properties of Dhvar-5-chitosan conjugates. The peptide grafted-chitosan showed significant reduced antimicrobial activity in solution, against all bacterial species tested, independently of peptide orientation and concentration. In order to further study the effect of covalently immobilized Dhvar-5 on antibiofilm activity against a biofilm formed by *S. epidermidis*, showed an efficient reduction of the total biofilm biomass.

A key point to address is related to the potential of AMP NPs to inhibit the formation and/or to eradicate the preformed biofilms. For example, novel strategies for effective biofilm inhibition and/or eradication are highly demanded for the prevention of infections and consequent rejection of implantable biomaterials. Most studies report on AMPs which are more effective in inhibiting the early phases of biofilm development rather than in eradicating established biofilms. It is possible that prevention of biofilm growth is related to killing of planktonic bacteria prior to attachment, and to inhibition of bacteria adhesion, which represent critical initial steps in biofilm formation. On the contrary, eradication of preformed biofilms may be attributed to the ability to disaggregate the matrix of preformed biofilms, and to diffuse into the deep layers of the biofilm killing bacteria inside the biofilm. Not many studies report on eradication ability of AMPs. We recently reported an interesting self-assembled peptide nanostructure decorated on the surface with the Amp WMR which was able to essentially eradicate already formed biofilm and may represent a novel strategy and may provide an added value for therapeutic applications [53].

## 5. In Vivo Models for Biofilms

To broaden in vitro findings, it is important to enforce in vivo mimicking conditions. Models of infection in vivo and pathogenesis are a continuous interaction between the host and microbes and between microbes themselves [67]. These interactions are complex and dynamic and make it difficult to study them in in vitro models.

To validate results in vitro, that could be further translated into higher organisms or clinical trials, it is necessary to use the right in vivo model [68]. It is very difficult to reproduce the host environment in in vitro models and for this reason there is a need to use multiple in vivo models ranging from non-mammalian to mammalian models, mainly rodents [69]. Because of ethical concerns, the use of in vivo models is now submitted to legal regulations, as it is necessary to consider not only medical benefits but also animal welfare [70]. This partly explains why researchers are still using in vitro and in vivo non-mammalian models. Non-mammalian models show advantages such as rapid development and cheap experiments to perform. Moreover, the genome of most of these organisms have been sequenced, and it is possible to do genetics both on the bacteria and the host. However, these non-mammalian models may have limitations such as the growth temperature of pathogens and are not suitable for studying chronic infections. In the past years, non-mammalian models like *Drosophila melanogaster*, among invertebrates, and *Danio rerio*, among vertebrates, have been adapted to study host-microbe interactions and immune system responses related to colonization of the intestine by biofilms [69].

Zebrafish are an emergent model to study human disease. They have a highly developed innate cellular immune systems that make them a perfect model to study infectious diseases. Currently, zebrafish are used as a model for imaging and genetics to investigate the molecular and cellular foundations of host–microbe relationships for the in vivo biofilm studies. Microbiome studies in zebrafish hosts have concentrated on embryonic and larval stages, when the advantages of this model are maximized. It is fundamental to understand the interactions between microbe and host to know the host developmental level when microbes colonize it and know the effects on host ontogeny [71].

Neely et al. [72] have produced a model of bacterial pathogenesis injecting *Streptococcus iniae* into zebrafish dorsal muscles. Zebrafish are also susceptible to *Streptococcus* pyogenes, the human pathogen. They used genetics and mutagenesis to obtain several attenuated mutants in zebrafish. The success of the combination Streptococcus–zebrafish makes it a great model for the analysis of infectious disease. Rawls et al. [73] colonized germ-free zebrafish with *P. aeruginosa* strains containing mutated genes for motility obtaining the attenuation of the host innate immune responses. They show the utility of germ-free zebrafish in determining the comportment and localization of bacteria in vertebrate intestine and identified the bacterial genes involved on host/microbial interactions. The protective mechanism against pathogens of probiotic or commensal bacteria in germ-free animal models, remain poorly understood. Rendueles et al. [74] pre-colonized zebrafish larvae with 37 probiotic bacterial strains to screen the survival upon *Edwardsiella ictaluri* infection identifying 3 strong protective strains, including *Vibrio parahaemolyticus* and two *E. coli* strains. They favor the emergence of probiotic bacteria in zebrafish larvae. The trait of tuberculosis is the formation of granulomas, Stoop et al. [75] identified new mycobacterial factors involved in the granuloma formation. They utilized the zebrafish embryo-Mycobacterium marinum model to selected *M. marinum* mutants that did not induce granuloma formation. One of the mutants was found to be defective in the espL gene, which is located in the ESX-1 cluster. This cluster encodes a specialized secretion system important for granuloma formation and virulence; it is modified in the BCG vaccine strain of *Mycobacterium bovis*. They concluded that this screen is useful to identify the genes involved in the initial stages of granuloma formation and to develop new tuberculosis vaccines.

Amphibian skin continuously comes in contact with diverse aquatic and/or terrestrial microbial environment and constitute the first line of defense against environmental pathogens working as an immune organ. The immune functions of frog skin are the maintenance of structural, physiological, and microbiological barriers; moreover, it permits the interactions across all these barriers. The cellular and molecular mechanisms that lead to the immune function and defense against pathogens of amphibian skin are still unknown. To date amphibians have not been considered to study in vivo biofilms because of their potent innate immuno-defense but, recently, amphibian immune responses and the contribution of commensal microbes to pathogen defense have been altered, probably due to environmental changes derived from natural and human factors and new pathogens infections [76]. In fact, pathogens that normally infect only mammals and humans have been identified in amphibians. They have caused serious damage to these vertebrates, even death. In 2016, epidemic meningitis-like disease due to an infection with *Elizabethkingia miricola* hit *Pelophylax nigromaculatus* an edible amphibian in south-central China. *Elizabethkingia* is a strains of Gram-negative, non-motile, non–spore-forming bacilli that causes bacteremia and septicemia in immunocompromised and immunocompetent patients [77]. Mühldorfer et al. reviewed the role of “unusual” *Brucella* in amphibians [78] and Ikuta et al. [79] discussed of the first episode of bovine tuberculosis in the amphibian *Lithobates catesbeiana*. Previously only nontuberculous mycobacteria infections were detected in amphibians. Environmental pollutants can increase disease risk in wildlife altering the interaction intestine-microbiota of hosts [80]. Research paradigms from humans, other animals, and plants show a great influence of environmental microbial ecosystems on the microbiota and health of organisms. They indicate that a strong link between environmental and internal microbial diversity and good health exists [81]. These considerations combined with the discovery that pathogens, previously considered exclusive to mammals, have also been found in amphibians, can push to consider, in the near future, the use of amphibians as a model for the study of in vivo biofilm. Furthermore, it could be useful to test the use of AMPs to contrast the diseases derived by these new amphibian pathogens and then transfer the data obtained to mammals, including humans. The amphibians, such as *Xenopus* genus, have a high genomic homology with humans and their genome has been completely sequenced, in particular *X. tropicalis*, has a short generation time and a diploid genome, which make it a good model for genetic studies [82]. Moreover, the development pathways of embryo are similar to those of mammals, including humans [83]. For these features they are considered, in the scientific world, an excellent and easy model to replace the use of mammals. Furthermore, it must be considered that amphibians diverged more recently from amniotes (360 million years ago) than fish (over 400 million years ago) and thus they could replace zebrafish another non-mammal model widely used in recent years for biofilm studies. Finally *Xenopus* can be easily bred in captivity, protocols for its laboratory breeding are simpler and of a lower cost than those required by rodents.

## 6. Conclusions

The antibiotic resistant pathogens are increasing and the capacity of new antimicrobial compounds to control bacterial infections is diminishing. Antimicrobial peptides represent an alternative to classical antibiotics to control and fight bacterial infections. This review analyses the potentially most suited features of peptides to combat not only bacteria in their planktonic mode of life but also effective against already formed biofilms (eradication). In vivo studies could be performed to enhance the activity of antimicrobial peptides to treat infections caused by multidrug-resistant Gram-positive and Gram-negative bacteria and the development of suitable models is extremely important. Clearly, strategies to improve protease susceptibilities of AMPs while still attaining an active nanosystems with enhanced proteolytic resistance constitute an added value for therapeutic applications. NPs can be exploited to favor this proteolic stability and produce supramolecular peptide-based platforms with potent antibacterial activity, improved functionalities, and enhanced biocompatibility compared to other antimicrobial molecules. Peptide-based nanosystems could serve also as an important class of novel antibacterial compounds for the treatment of intracellular microbial infections. To further improve antibiofilm properties, AMP NPs can be also combined with conventional antibiotics and allow to decrease the effective concentration of the active molecules, as well as to extend their spectrum of action, thereby reducing the spread of resistance, which is often linked to monotherapy regimens.

Promisingly, in vivo studies often showed effects that significantly improved clearance of bacterial isolates from the infection site, regardless of the peptides mode of action including enhancement of penetration and potential disruption of the stringent stress response.

Thus, it remains a major challenge to translate in vitro findings into in vivo efficacy because often compounds that show excellent in vitro activity work poorly when tested under in vivo conditions.

## Figures and Tables

**Figure 1 pharmaceutics-11-00322-f001:**
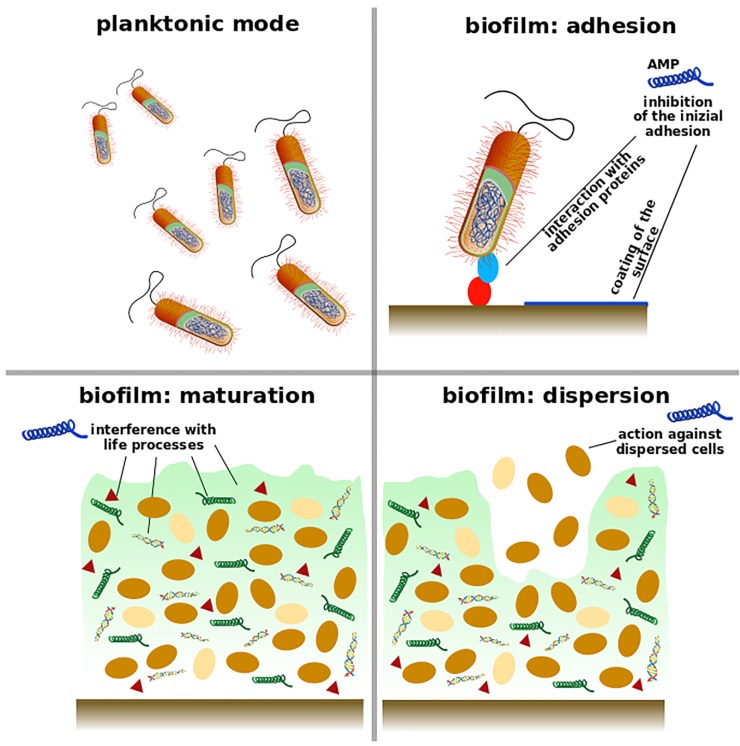
Biofilms develop in three phases: initial adhesion, maturation, and dispersion. A good anti-biofilm agent (blue peptide) can be involved in all the biofilm phases. AMPs may inhibit the accumulation of bacteria on the surface by interacting with their adhesion proteins (red and light blue) or coating the surface to protect from bacterial attack. Antimicrobial peptides can display an action against bacteria in their active state in the biofilm (brown); they can also defeat bacteria in the persisters models (yellow) or interfere with life processes such as synthesis of EPS (light green matrix), signaling compounds (triangle), extracellular DNA, and proteins (green helix).

**Figure 2 pharmaceutics-11-00322-f002:**
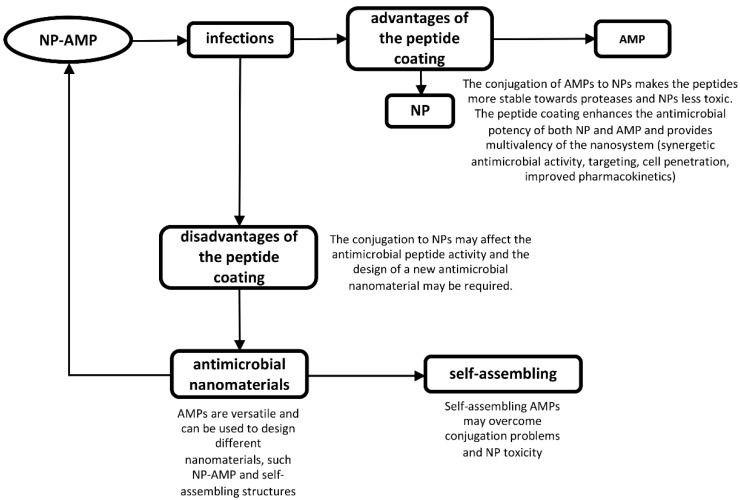
Coating of nanoparticles with antimicrobial peptides is a promising strategy to overcome the drawbacks of the nanoparticles and peptides themselves and improve their action with a synergistic effect. Moreover, the conjugation may affect the peptide properties and the design of new antimicrobial nanomaterials is required.

**Table 1 pharmaceutics-11-00322-t001:** Schematic representation of some AMP extracted by amphibian.

Amphibian AMPs Production				
Species	Family	Peptide	Effects	Pathogens
*X. laevis*	pipidae	XLAsp-P1	haemolytic activity, destruction of the membrane	Gram-positive and Gram-negative
*L. infrafrenata*	Hylidae	frenatin 4.1 frenatin 4.2. frenatin 4.2a	antimicrobial activities	all bacteria tested
*P. hypochondrialis*	Hylidae	Dermaseptin-PH	inhibition of the growth	Gram-negative and Gram-positive *Candida albicans*
*P. tarsius* *P. hypochondrialis*	Hylidae	phylloseptin-PTa and phylloseptin-PHa	antimicrobial activities	*S. aureus*, *E. faecalis*
*P. kl. esculentus*	Ranidae	temporin-PE	antimicrobial activities	antimicrobial activities
*A. davidianus*	Cryptobranchidae	andricin B	antimicrobial activities	all bacteria tested and some fungi *Staphylococcus aureus*
*L. fujianensis*	Dicroglossidae	Japonicin-2LF	membrane permeabilization	*S. aureus* and MRSA

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
