# Peer review of "Biofilms: Novel Strategies Based on Antimicrobial Peptides"

_pharmaceutics, 2019, doi:10.3390/pharmaceutics11070322_

Reviewer 1 Report

The manuscript by Galdiero et al. reports a review on some recent strategies in the development of antimicrobial peptides against biofilms. In the review, the authors first outlined the necessity of developing novel antimicrobial therapies and then described the properties of biofilm followed by strategies that could be adopted against biofilm. This is followed by a review on the anti-biofilm effects of different AMPs, and the use of nanoparticles to enhance the stability and activity of AMPs. Towards the end of the manuscript, the authors also mentioned the use of animal models and the associated difficulties in studying the in vivo properties of AMPs, and how these could be overcome by the use zebra fish and amphibian models.

While this manuscript provides an informative review on AMPs against biofilms, it requires a very thorough proofread. The introductory section is quite repetitive and contains general information that is well known. There are numerous typos in the manuscript and many sentences require rephrasing.  A selection of these is listed below:

Line 39 – “is not the only responsible”

Line 40 – proctices

Line 61 - lipid, complement, fibronectin

Line 73 – what is LES

Line 79 - C.parapsilosis,

Line 111 - enzymes synthesis.

Line 131 – 135 – long sentence

Line 136 - have proved to play an action against

Line 137 - have shown ability to

Line 146 - (brown spots)??

Line 147 – Yellow??

Line 151 - desired property for anti-biofilm agents

Line169 – delete “of the last century”

Line 170 – skyn

Line 170 - AMPs are constantly isolated

Line174 - showed in vitro  high activity

Line 176 - acts destroying

Line 178 - identified the peptides: frenatin 4.1

Line 207 - they act preventing the

Line 208 - pathways as colonization

Line 216 - and maintaining hydrated the biofilm environment

Reference #51 seems incomplete.

Author Response

Comments and Suggestions for Authors 1

The manuscript by Galdiero et al. reports a review on some recent strategies in the development of antimicrobial peptides against biofilms. In the review, the authors first outlined the necessity of developing novel antimicrobial therapies and then described the properties of biofilm followed by strategies that could be adopted against biofilm. This is followed by a review on the anti-biofilm effects of different AMPs, and the use of nanoparticles to enhance the stability and activity of AMPs. Towards the end of the manuscript, the authors also mentioned the use of animal models and the associated difficulties in studying the in vivo properties of AMPs, and how these could be overcome by the use zebra fish and amphibian models.

While this manuscript provides an informative review on AMPs against biofilms, it requires a very thorough proofread. The introductory section is quite repetitive and contains general information that is well known. There are numerous typos in the manuscript and many sentences require rephrasing.  A selection of these is listed below:

The introduction was revised

Line 39 – “is not the only responsible We have modified accordingly

Line 40 – proctices  We have modified accordingly

Line 61 - lipid, complement, fibronectin We have modified accordingly

Line 73 – what is LES We have modified accordingly

Line 79 - C.parapsilosis, We have modified accordingly

Line 111 - enzymes synthesis.  We have modified accordingly

Line 131 – 135 – long sentenceWe have modified accordingly

Line 136 - have proved to play an action against We have modified accordingly

Line 137 - have shown ability to We have modified accordingly

Line 146 - (brown spots)?? We have modified accordingly

Line 147 – Yellow?? We have modified accordingly

Line 151 - desired property for anti-biofilm agents We have modified accordingly

Line169 – delete “of the last century” We have modified accordingly

Line 170 – skyn We have modified accordingly

Line 170 - AMPs are constantly isolated We have modified accordingly

Line174 - showed in vitro  high activity We have modified accordingly

Line 176 - acts destroying We have modified accordingly

Line 178 - identified the peptides: frenatin 4.1 We have modified accordingly

Line 207 - they act preventing the We have modified accordingly

Line 208 - pathways as colonization We have modified accordingly

Line 216 - and maintaining hydrated the biofilm environment We have modified accordingly

Reference #51 seems incomplete. The reference is complete

Reviewer 2 Report

Biofilms are difficult to treat so a comprehensive review on this topic is welcomed. There are some major issues that need to be addressed:

1) explain why AMPs should be considered instead of other small molecules with potent anti-biofilm activities, despite the many challenges of AMP.

2) The review did not distinguish biofilm formation inhibition from biofilm eradication. Can these AMP cause biofilm eradication? Can nanoparticle AMP eradicate established biofilms? If they only prevent biofilm formation, that is not exciting.

3) What is the mechanism of AMP that inhibit biofilm formation at sub MIC? Merely saying they reduce c-di-GMP is not enough. How do they reduce c-di-GMP? Known?

4) Provide a better perspective of the field. What innovation needs to be done for AMP as antibiofilm agents to get into clinic?

5) In nature, there are hardly monospecies biofilms. Discuss work done on polymicrobial biofilms.

Other comments: English and grammar is poor at many places. Some spelling errors noted.

Author Response

Comments and Suggestions for Authors 2

Biofilms are difficult to treat so a comprehensive review on this topic is welcomed. There are some major issues that need to be addressed:

1)     explain why AMPs should be considered instead of other small molecules with potent anti-biofilm activities, despite the many challenges of AMP.

The increasing emergence of multidrug resistant bacteria is a serious threat to public health, there is an urgent need to develop new antimicrobial agents that are active against bacteria and less likely to induce drug resistance. AMPs are attractive candidates as therapeutic agents, because they possess antimicrobial, anti-inflammatory and anti-biofilm activities, but also possess the ability to bind LPS and block LPS-stimulated cytokine release(Cationic host defence peptides: innate immune regulatory peptides as a novel approach for treating infections Cell. Mol. Life Sci., 64 (2007), pp. 922-933). Moreover their weaknesses such as low cell selectivity and high production costs can be overcome with a good peptide design. Combination between small molecules and AMPs may represent a strategy.

2)     The review did not distinguish biofilm formation inhibition from biofilm eradication. Can these AMP cause biofilm eradication? Can nanoparticle AMP eradicate established biofilms? If they only prevent biofilm formation, that is not exciting. We agree with the referee that eradication is key for medical applications. Both processes are discussed into the review from a general point of view; now we added a paragraph specifically dealing with this difference and explaining that only a few examples of eradicating systems are reported in literature. Notwithstanding this, many new studies are continuously reported in literature.

3) What is the mechanism of AMP that inhibit biofilm formation at sub MIC? Merely saying they reduce c-di-GMP is not enough. How do they reduce c-di-GMP? Known? Unfortunately this is not known and we hypothesized that they could perturb the life-style of biofilms

4) Provide a better perspective of the field. What innovation needs to be done for AMP as antibiofilm agents to get into clinic? In vivo studies are critical to show the correspondence with the in vitro results for future clinical applications. Certainly, innovation comprise the greater proteolytic stability, bioavailability, biodegradability and possibility of using combination strategies. We now addressed clearly this point in the conclusion paragraph.

5) In nature, there are hardly monospecies biofilms. Discuss work done on polymicrobial biofilms. We agree with the referee that this is an important point and indeed we already described in the review the polymicorbial biofilms. We modified the text so to make this more clear.

Other comments: English and grammar is poor at many places. Some spelling errors noted.

We now improved English and checked the manuscript for spelling errors.

Reviewer 3 Report

This review by Galdiero et al provides a nice overview of the relatively recent field of anti-biofilm peptides and should be of interest to the readership of this journal.

Minor comments:

- please improve language throughout the manuscript.

- it would be nice to generate a table or figure showing the key peptides and their corresponding sequence requirements responsible for anti-biofilm properties.

- Fig. 2. Although I agree that functionalizing peptides with nanoparticles is promising, what is the current evidence suggesting this is a successful approach?

- Is there evidence that anti-biofilm peptides operate by binding and degrading eDNA?

- Add some of the pioneering papers of this field and discuss them in the text. Examples: doi: 10.1128/AAC.00064-12

- Generating a table outlining animal models for anti-biofilm peptide research would add to the review

Author Response

This review by Galdiero et al provides a nice overview of the relatively recent field of anti-biofilm peptides and should be of interest to the readership of this journal.

Minor comments:

- please improve language throughout the manuscript. English was carefully revised

- it would be nice to generate a table or figure showing the key peptides and their corresponding sequence requirements responsible for anti-biofilm properties. We thank the reviewer for this suggestion, we believe that this table/figure would not provide added values to the present paper but would be of great interest for a review paper focussed on database generation.

- Fig. 2. Although I agree that functionalizing peptides with nanoparticles is promising, what is the current evidence suggesting this is a successful approach? There are many evidences reported in literature about results on peptides and nanoparticles evidencing that this combination can reduce toxicity issues and enhance activities. Some of this are reported in references from 48 to 52.

- Is there evidence that anti-biofilm peptides operate by binding and degrading eDNA?

To the best of our knowledge, there are no studies about this topic, but it was shown that oligosaccharides could interact with e-DNA of Pseudomonal biofilm. (see Targeted disruption of the extracellular polymeric network of Pseudomonas aeruginosa biofilms by alginate oligosaccharides.L. C. Powell1, M. F. Pritchard, E. L. Ferguson, K. A. Powell1, S. U. Patel1, P. D. Rye, S.-M. Sakellakou,N. J. Buurma, C. D. Brilliant, J. M. Copping, G. E. Menzies , P. D. Lewis, K. E. Hill and D. W. Thomas. NPJ Biofilms Microbiomes. 2018 Jun 29;4:13. doi: 10.1038/s41522-018-0056-3. eCollection 2018.)

- Add some of the pioneering papers of this field and discuss them in the text. Examples: doi: 10.1128/AAC.00064-12 We thank the referee for his suggestion and we added the citation

- Generating a table outlining animal models for anti-biofilm peptide research would add to the review  We thank the referee for this suggestion and we added the Table

Round  2

Reviewer 1 Report

The authors have addressed the comments and suggestions from the last review.

Reviewer 2 Report

revised manuscript is better and improved.

Acceptance recommended

Pharmaceutics EISSN 1999-4923 Published by MDPI AG, Basel, Switzerland RSS E-Mail Table of Contents Alert
Back to Top